# Communicating Evidence about the Causes of Obesity and Support for Obesity Policies: Two Population-Based Survey Experiments

**DOI:** 10.3390/ijerph17186539

**Published:** 2020-09-08

**Authors:** James P. Reynolds, Milica Vasiljevic, Mark Pilling, Marissa G. Hall, Kurt M. Ribisl, Theresa M. Marteau

**Affiliations:** 1Behaviour and Health Research Unit, University of Cambridge, Cambridge CB2 0SR, UK; milica.vasiljevic@durham.ac.uk (M.V.); mark.pilling@medschl.cam.ac.uk (M.P.); 2Department of Psychology, Durham University, Durham DH1 3LE, UK; 3Lineberger Comprehensive Cancer Center, University of North Carolina at Chapel Hill, Chapel Hill, NC 27514, USA; mghall@unc.edu (M.G.H.); kurt_ribisl@unc.edu (K.M.R.); 4Department of Health Behaviour, UNC Gillings School of Global Public Health, University of North Carolina at Chapel Hill, Chapel Hill, NC 275599, USA

**Keywords:** policy, acceptability, overweight, attributions, framing, communication

## Abstract

Public support for numerous obesity policies is low, which is one barrier to their implementation. One reason for this low support is the tendency to ascribe obesity to failings of willpower as opposed to the environment. Correlational evidence supports this position. However, the experimental evidence is mixed. In two experimental studies, participants were randomised to receive no message, messages about the environment’s influence on obesity (Study 1 & 2), or messages about the environment’s influence on human behaviour (Study 1). We investigated whether communicating these messages changed support for obesity policies and beliefs about the causes of obesity. Participants were recruited from nationally representative samples in Great Britain (Study 1 & 2) and the USA (Study 2) (total *n* = 4391). Study 2 was designed to replicate existing research. Neither study found evidence that communicating the messages increased support for obesity policies or strengthened beliefs about the environment’s role in obesity. Study 2, therefore, did not replicate two earlier experimental studies. Instead, the studies reported here suggest that people’s beliefs about the causes of obesity are resistant to change in response to evidence and are, therefore, not a promising avenue to increase support for obesity policies.

## 1. Background

Obesity rates are high and rising worldwide, which is leading to higher rates of type 2 diabetes and numerous cancers [1,2]. Several policies could reduce and prevent obesity, including those that change obesogenic environments. However, there are a number of barriers to getting these policies passed into law [3]. Lack of public support for many of the proposed policies [4,5,6,7] is identified as one major cause of this inertia, alongside inadequate political leadership and strong opposition from powerful commercial interests [3]. There are numerous reasons for low public support, including the public’s beliefs about the causes of obesity, the focus of this paper.

Beliefs about the causes of phenomena—causal beliefs—are core to attribution theory, which describes how people form explanations for their own and others’ behaviour [8,9]. Evidence from multiple sources demonstrates that people spontaneously generate causal explanations for phenomena [10,11]. Two sets of causal beliefs concern the self (e.g., personal characteristics such as willpower; internal attributions), and the situation (e.g., the presence of others in a social setting [9]; external attributions). As Kelley ([9]; p.107) puts it: “If a person is aggressively competitive in his behavior, is he this kind of person, or is he reacting to situational pressures?”. The formation of causal beliefs is often based on implicit assumptions and incomplete data, and thus prone to bias [12]. One such bias is the correspondence bias, or fundamental attribution error, whereby people overestimate the role of the self in the causes of behaviour and discount situational influences [13] (e.g., if someone is late to work, a co-worker may infer that the person is lazy and discount situational explanations such as disruption to transport systems or caring responsibilities). There is also some evidence for a self-serving bias whereby people are more likely to attribute a negative behaviour to personal characteristics when they are observing others engage in this behaviour and less likely to attribute this same behaviour to personal characteristics when they themselves are engaged in this behaviour [14]. Given the self-protective effects of causal attributions, it is unsurprising that they may, at best, only be a rough approximation of reality.

Research based on attribution theory has led to the well-replicated result that causal beliefs are associated with attitudes. For example, the belief that homosexuality is a choice is associated with negative attitudes toward people who are homosexual and with less support for equal rights and same-sex marriage [15,16,17]. The belief that people who are poor are responsible for their own misfortune is associated with negative attitudes and less support for policy measures for those who are poor [18,19]. Similar results are reported for other groups: people who are transgender, people with a mental illness, people with criminal records, and, most relevant to this paper, people who are obese [4,20,21,22,23].

Most people believe that being overweight or obese is due to a lack of personal responsibility, with fewer people acknowledging the influence of the environment, such as the widespread availability of unhealthy foods and lack of public space for physical activity [4,5,24,25,26]. This may indicate the existence of the correspondence bias for obesity. Exposure to media coverage of obesity may further entrench these beliefs as newspapers are more likely to highlight individual-level drivers for obesity, an effect that is magnified in newspapers categorised as less liberal [27,28,29]. This overestimation of the role of individual responsibility at the expense of environmental influences may explain, in part, the relatively low public support for government intervention to tackle obesity by changing environments [4,24,30]. As this past research is correlational, it is difficult to determine if these beliefs are consequential, i.e., whether believing that the environment causes obesity leads to more favourable attitudes toward policies that help to reduce obesity.

Several studies have attempted to increase support for obesity policies by drawing on attribution theory and communicating information about the environment’s influence on obesity. For example, one study [31] communicated a message that included statements highlighting how the high availability and low price of unhealthy foods contribute toward obesity. Participants who read these messages reported greater support for policies to reduce obesity compared to participants in the control group who did not read any message. Subsequent studies have either found smaller effect sizes [32] or have not replicated these effects [33,34,35]. A further study only found statistically significant effects in a subgroup; among male but not female participants [36]. It is unlikely that differences in intervention content explain these mixed effects. Although the interventions varied across studies, they shared several key messages about the influences on obesity, including portion size [31,36], the availability of unhealthy foods [31,33,34,35,36], advertising of unhealthy foods [31,32,36], and the lower price of unhealthy foods [31,34,35,36].

Given the mixed results from the experimental literature, which have become apparent amid concerns about the reproducibility of existing research [37], robust studies are needed to reduce the existing uncertainty. In addition to the specific approach of changing obesity attributions to influence support for policies to reduce obesity, we also investigated whether people’s broader attributions about human behaviour may influence support for obesity policies that aim to change behaviour.

## 2. Study 1

The aim of Study 1 was to identify messages that would be most effective at changing causal beliefs for use in Study 2. A further aim was to investigate whether and to what extent these messages changed support for obesity policies.

### 2.1. Method

This study was pre-registered with the Open Science Framework (DOI: [38]). There was one deviation from the registered protocol: we increased the sample size from 375 to 1681 to increase statistical power, in line with a recent study with similar methods [32]. This decision was made after the protocol was registered, but before commencement of data collection. Supporting data and the full questionnaire can also be found in the OSF folder.

The Cambridge Psychology Research Ethics Committee based at the University of Cambridge approved both studies on 13 July 2018 (No. PRE.2019.006). Consent was provided by all participants digitally prior to completing the online questionnaire. The study was conducted in accordance with the Declaration of Helsinki.

#### 2.1.1. Participants

A nationally representative sample from Great Britain (*n* = 1681) was recruited via YouGov’s online panel (www.yougov.co.uk). The recruitment method used quotas for age, gender, social grade, education, region, political attention, and voting in the 2017 General Election and 2016 EU referendum. Data were collected between 29th and 30th August 2018. After applying sample weights that were provided by the research agency to ensure the representativeness of the sample, mean age = 48.33 (SD = 16.87) and 51.6% were female. See Appendix A for the full demographic characteristics of the sample.

This sample size ensured similar group sizes from a comparable study [32]. The exact effect size from Ortiz, Zimmerman and Adler [32] could not be calculated as the descriptive statistics were not reported in the manuscript; however, a sample size calculation suggested that this sample size would provide 80% power to detect small effects between two groups (Cohen’s *d* = 0.26) after a Bonferroni adjustment (*α* = 0.0125). The Gpower software v3.1 [39] was used to conduct the sample size calculation. The test family was t tests, the statistical tests was difference between two independent means, the allocation ratio was set at 1, and a two tailed test was selected.

#### 2.1.2. Design

The study was an online, between-subjects experiment in which participants were randomly allocated to one of five groups differing in the messages that they received about (i) the environment’s influence on obesity, or (ii) the environment’s influence on human behaviour in general:

Group 1: Control group (no message).

Group 2: Obesity version (a): received a message that highlighted the role of food availability and cost.

Group 3: Obesity version (b): received a message that highlighted the role of cost, availability, and marketing.

Group 4: Behaviour version (a): received a message that highlighted the role of availability and cost in shaping behaviour in general.

Group 5: Behaviour version (b): received a message that highlighted the role of availability, cost, advertising, and portion size in shaping behaviour in general.

After viewing the randomly assigned messages, participants completed a short questionnaire. The randomisation and questionnaire were programmed in YouGov’s software.

#### 2.1.3. The Interventions

The interventions comprised four messages (see Appendix A). We hypothesised that the two obesity messages (Groups 2 & 3) would strengthen the belief that the physical environment causes obesity when compared to the control group (Group 1); and that the two behaviour messages (Groups 4 & 5) would strengthen the belief that the physical environment influences human behaviour more generally when compared to the control group (Group 1). No directional predictions were made over which message would be most effective at changing the target belief or public support. It was planned that one message from each of the two sets—obesity and human behaviour—would be selected for use in Study 2 based on their effectiveness at instilling or strengthening the target causal belief and on the self-reported subjective comprehension of the message’s content.

The message used in Group 2 was adapted from Pearl and Lebowitz [31], with images of the obesogenic environment added below the text as the addition of images to text has been shown to increase attention, comprehension, and recall of information [40]. These images included examples of the message content, such as the high availability of less healthy foods and aggressive food advertising. The obesity message presented in Group 3 was developed specifically for the present study based on aspects of the environment that have been linked with obesity: cost (less healthy food is cheaper), availability (less healthy food is widely available), and marketing (less healthy food is heavily advertised) [41,42]. References to “evidence” and “research” were added to increase the persuasiveness of the message [43,44,45]. The messages in Groups 4 and 5 were designed to mimic the structure of the two obesity messages, however the focus was changed from obesity to human behaviour in general, in which two examples are given: which mode of transport people use and whether people purchase items in single-use plastics. These examples were chosen as they are daily behaviours that are influenced by the same environmental factors of cost, availability, and marketing.

#### 2.1.4. Measures 

##### Policy Support

Support for three obesity prevention policies was assessed using a single item adapted from earlier research [5]: “Do you support or oppose the new policy?” rated on a seven-point scale (1 = Strongly oppose; 7 = Strongly support). The three policies were: a 20% tax on confectionary, reduction in the size of unhealthy ready meals, and banning advertising for unhealthy foods during children’s television. The three policies were presented as a package, so participants rated whether they supported or opposed the implementation of all three. These three policies were chosen as they have not been implemented in the UK, and research suggests that they would be effective [46,47,48].

##### Causal Beliefs (Manipulation Checks)

The belief that obesity is influenced by the food environment (Obesity Causal Beliefs: Environment) was measured with two items (*r* = 0.71) that were adapted from previous research [5,30]: “The low cost, widespread availability and marketing of unhealthy foods are to blame for the high rates of obesity” and “People are obese because there are so many cheap, unhealthy foods around”. The belief that human behaviour is influenced by the environment (Behavioural Causal Beliefs: Environment) was measured with two items (*r* = 0.56) that were adapted from the Obesity Attributions items described above: “People’s behaviour is strongly influenced by their environment and surroundings” and “The cost and availability of products influence what people buy and choose”. The causal belief items were rated on a seven-point scale (1 = Strongly disagree; 7 = Strongly agree) and were presented in counterbalanced order. These questions were given to all participants, regardless of intervention group.

##### Comprehension

Participants rated the clarity and comprehension of the intervention that they were randomised to read (Subjective Comprehension) with two items (*r* = 0.78): “I found the information in the summary I just read clear” and “I found the information in the summary I just read easy to understand”. The items were rated on a seven-point scale (1 = Strongly disagree; 7 = Strongly agree) and were presented in counterbalanced order.

##### Other Variables

The research agency provided demographic data including age, gender, socio-economic status [49], education [50], and region. Educational achievement was recoded into three categories: low education (no education, GCSEs (General Certificate of Secondary Education) or similar); medium education (A-levels, non-degree teaching qualifications, or similar); and, high education (degree awards or higher). Socio-economic status was also recoded into three categories: low (DE), medium (C1C2), and high (AB). The recoding was done in accordance with previous research [6]. The variable region was not recoded, and the specific regions included in the model were those provided to us by the research agency.

#### 2.1.5. Analyses

To determine whether the groups were matched on key variables following the randomisation we used the percentage method to detect chance imbalances [51]. Potential confounding variables (gender, age, SES, education, and region) were compared across groups and if there were differences beyond 5% points (e.g., Group 1: 47% female; Group 2: 53% female) it was concluded that there were chance imbalances. Several chance imbalances above 5% points were identified for all five variables across the groups. Thus, the main analyses used ordinary least squares regression, controlling for these five demographic characteristics, to test the main effects of experimental group on support for policies to tackle obesity and beliefs about the causes of obesity. Sensitivity analyses were conducted in which covariates were not included, to determine whether the main pattern of results would change (see Appendix A). Model diagnostics (residual plot, Normal p-p plot of residuals) were examined and showed that the regression modelling assumptions were satisfied.

The criteria for significance was set at *α* = 0.0125 for all four outcomes (*α* = 0.05/4 = 0.0125), after applying a Bonferroni adjustment. Outliers (±3SDs from the mean) on continuous variables were removed. 30 outliers were removed (2%) from the Behavioural Causal Beliefs: Environment variable, and 31 were removed from the Subjective Comprehension variable (2%). There were no other outliers. Sensitivity analyses were conducted in which outliers were not excluded, to determine whether the main pattern of results changed. Cohen’s *d* statistics are covariate-adjusted.

### 2.2. Results

#### 2.2.1. Policy Support

None of the intervention messages increased support for the obesity prevention policies when compared to the control group (all *p*s > 0.0125, see Table 1 for full results).

#### 2.2.2. Causal Beliefs (Manipulation Checks)

There was a statistically significant effect of the message presented in Group 2 pertaining to the environment’s role in obesity on beliefs about the environment’s influence on human behaviour. Participants who received this intervention message believed that the environment had a greater influence on human behaviour than those in the control group, *B* = 0.18, 95% CIs [0.04, 0.31], *p* = 0.009, *d* = 0.20, representing a small increase on the 1–7 rating scale (see Table 1 for full results).

There was also a statistically significant effect of Group 3’s message about the environment’s influence on obesity on beliefs about the environment’s influence on obesity. Participants who received this intervention believed that the environment had less influence on obesity than those in the control group, *B* = −0.27, 95% CIs [−0.47, −0.07], *p* = 0.007, *d* = 0.21. This effect was in the opposite direction to that which was predicted. 

There were no other statistically significant effects of any intervention, compared to the control group, on beliefs about the environment’s influence in obesity or in human behaviour in general (all *p*s > 0.0125).

#### 2.2.3. Subjective Comprehension

There were no statistically significant differences between the two obesity messages in terms of subjective comprehension, *B* = −0.04, 95% CIs [−0.19, 0.11], *p* = 0.623, *d* = 0.04. However, the human behaviour message communicated in Group 5 was rated as significantly clearer and easier to understand when compared to the human behaviour message in Group 4, *B* = 0.21, 95% CIs [0.07, 0.36], *p* = 0.005, *d* = 0.22, a small increase on the 1–7 rating scale (see Appendix A for descriptive statistics).

### 2.3. Discussion

The primary aim of this study was to select messages that would be most effective at changing causal beliefs for use in Study 2. The results showed that none of the interventions changed the target belief in the hypothesised direction. In keeping with this, there were no changes in support for obesity-related policies. However, we found that participants who read one of the messages containing information about the environment’s influence on obesity (Group 2) were more likely to believe that the environment influenced human behaviour than those in the control group. Also, those who read the second obesity message (Group 3) were less likely to believe that the environment influenced obesity than those in the control group. Despite changes in causal beliefs from those who read these two messages, there was no subsequent change in support for policies among these participants. These findings suggest that messages designed to induce the belief about the environment’s influence on both obesity-related behaviours and human behaviour, in general, do not directly influence attitudes toward obesity policies.

While the results from existing studies have been mixed, the lack of evidence for an effect of Group 2’s message, in particular, was unexpected as this message had previously been found to increase public support in another study [31]. Other studies using similar interventions also showed changes in attitudes toward obesity policies and beliefs about the causes of obesity [32]. We offer two possible explanations about the conflicting results between our study and those of two others that reported statistically significant effects [31,32]. The first explanation concerns cultural differences. Studies conducted by [31,32] were both conducted in the USA. The current study was conducted in Great Britain. There is some evidence that US populations are less supportive of regulation to change health-related behaviour than are those in Great Britain which may affect the sensitivity of the two populations to messages targeting these attitudes [52,53]. Second, there may be other differences in sample characteristics that affected responses to the interventions. The sample used in Pearl and Lebowitz [31] consisted solely of people who were overweight or obese. It is possible that people who are already overweight or obese may be more likely to revise their beliefs about the causes of obesity when presented with information about the environmental influence on obesity-related behaviours. However, this hypothesis remains untested.

## 3. Study 2

The results of Study 1 were originally intended to inform which interventions to use in Study 2. However, as none of the four interventions changed the target belief in the hypothesised direction and none changed public support, we decided to examine the replicability of the previous studies that successfully increase policy support, which we assumed would replicate prior to conducting Study 1. The revised aim of Study 2 was to determine if we could replicate previously published effects on the communication of obesity attribution messages and support for obesity policies. To do this, we decided to use two interventions used in two previously published studies reporting effects of communicating information about the environment’s influence on support for obesity policies [31,32]. We decided to reuse the intervention from Pearl and Lebowitz [31] that we tested in Study 1, but we removed the images to ensure that the presentation of the message was identical to the original study. The second message that we selected to test also successfully changed beliefs and attitudes in its original study [32]. A further goal was to test two explanations for the conflicting results between Study 1 and previous research: differences in nationality and BMI. To do this, we tested for interactions between intervention, country (England vs USA), and BMI. Based on prior research, Study 2 was designed to test three pre-registered hypotheses:ICommunicating messages that attribute obesity to the environment will (a) increase support for obesity prevention policies and (b) strengthen the belief that the environment causes obesityIIThese effects will be greater amongst:
participants from the USAparticipants who are obese or overweightIIIParticipants from England will report greater levels of support for obesity prevention policies and will be more likely to believe that the environment causes obesity.

### 3.1. Method

This study was pre-registered with the Open Science Framework [38]. There was one deviation from the registered protocol. The criteria for significance was changed as a principal components analysis (PCA) suggested a two-factor solution for our primary outcome. This change is described in the analysis section below. Supporting data and the full questionnaire can also be found in the OSF folder.

#### 3.1.1. Participants

Two nationally representative samples from England (*n* = 1397) and from the USA (*n* = 1315) were recruited via YouGov’s existing online panels. The recruitment method used quotas including age, gender, and education (for both countries); social grade, region, political attention, voting in the 2017 General Election and 2016 EU referendum race (for England only); and race, voter registration, and voting in the 2016 Presidential Election (for the USA only). These were different for the two countries as the research agency had different methods for ensuring a representative sample for each country. Data were collected between 10 and 13 December 2018. After applying sample weights provided by the research agency, the mean age = 48.36 (SD = 17.01) and 51.5% were female for the English sample and mean age = 47.31 (SD = 17.69) and 51.4% were female for the USA sample. See Appendix A for the full demographic characteristics of the sample.

This sample size ensured a similar group size from a comparable study [32] and approximately 30 times greater group size from another [31]. A sample size calculation suggested that the current sample size would provide 80% power to detect small effects between two groups (*d* = 0.14) after a Bonferroni adjustment (*α* = 0.025) and when combining the two samples. The Gpower software v3.1 [39] was used to conduct the sample size calculation. The test family was t tests, the statistical tests was difference between two independent means, the allocation ratio was set at 1, and a two tailed test was selected.

#### 3.1.2. Design

We conducted an online between-subjects experiment, in which participants were randomly allocated to one of three groups (see Table 2) differing in their exposure to messages about the environment’s influence on obesity.

Group 1: Control group: received no message.

Group 2: Obesity (a) Availability and cost: received a message that highlighted the role of food availability, cost, advertising, and portion size [31].

Group 3: Obesity (c) Advertising and placement: received a message that highlighted the role of food advertising and placement of unhealthy foods in supermarkets [32].

The randomisation was conducted using the research agency’s software. Participants completed a short questionnaire after receiving the interventions. The study was conducted simultaneously in England and the USA.

#### 3.1.3. The Interventions

Two messages were taken from previous studies. These messages below highlight several aspects of the environment that have been shown to influence obesity: cost, availability, portion size, placement, and marketing [41,42,54]. The only changes made were to the country name used (to match this to the two countries in which the current study was taking place). The full messages can be found in the Appendix A.

#### 3.1.4. Measures 

##### Primary Outcome(s)

Acceptability of seven policies, randomly ordered, was assessed using one response item for each [5]: “Do you support or oppose the new policy?” rated on a seven-point scale (1 = Strongly oppose; 7 = Strongly support). These seven policies were: a 20% tax on confectionary; reduction in the size of unhealthy snack foods; banning advertising for unhealthy foods during children’s television; a policy to increase the availability of healthy foods in worksites, schools, and hospitals; a limit on the maximum size of sugar-sweetened beverages in fast food restaurants; calorie labels on restaurant menus; and a ban on unhealthy snack foods in schools. We used a more comprehensive set of policies in Study 2 to match the policies assessed in the studies from which we sourced the interventions [31,32]. These seven items were converted into two outcomes: support for encouraging policies and support for discouraging policies (see Analyses section).

##### Causal Beliefs (Manipulation Checks)

The belief that obesity is caused by the food environment, genetics, and a lack of willpower were each measured with two response items (*r* = 0.73; 0.70; 0.77, respectively) [5]: “[Cause] is to blame for Obesity” and “People are obese because of [Cause]”. Each was rated on a seven-point scale (1 = Strongly disagree; 7 = Strongly agree). These items were presented in counterbalanced order.

##### Other Variables

BMI was calculated from self-reported height and weight. The research agency provided demographic data including age, gender, socio-economic status [49], education (Adapted from: [50]), and region. For the English sample, educational achievement was recoded into three categories: low education (no education, GCSEs or similar); medium education (A-levels, non-degree teaching qualifications, or similar); and, high education (degree awards or higher). Socio-economic status was also recoded into three categories: low (DE), medium (C1C2), and high (AB). For these transformations, see the methods section reported in Study 1. 

For the USA sample, educational achievement was recoded into four categories: low education (no high school, high school graduates), medium-low education (some college, 2 year college), medium-high education (4 year college graduate), and high education (post-graduate degree).

#### 3.1.5. Analyses

A PCA was conducted on the seven acceptability items with oblique rotation (direct oblimin). The Kaiser-Meyer-Olkin measure verified the sampling adequacy (KMO = 0.85; Bartlett’s test < 0.01), well above the minimum 0.50 that is needed [55]. Examining the scree plot and the Eigenvalues (>1) suggested a two-factor solution for the policy support items that explained 67% of the variance: (1) support for policies to discourage consumption of unhealthy foods and drinks (Discouraging policies); and (2) support for policies to encourage consumption of healthy foods (Encouraging policies). Support for Discouraging policies ranged from −2.07 (*strongly oppose*) to 2.13 (*strongly support*) and support for Encouraging policies ranged from −2.68 (*strongly oppose*) to 1.68 (*strongly support*). See Appendix A for factor loadings.

The main analyses used hierarchical OLS regressions to test the main effects and interactions between country, intervention group, and BMI on support for policies to tackle obesity and beliefs about the causes of obesity. The pre-registered criteria for significance was set at *α* = 0.05 for the primary outcome (policy support), and *α* = 0.05/4 = 0.0125 for the three secondary outcomes, after applying a Bonferroni multiplicity adjustment. However, as the PCA suggested a two-factor solution for policy support, this was changed to *α* = 0.025 for the co-primary outcomes and *α* = 0.01 for the three secondary outcomes.

Potential confounding variables (SES, education, gender, age, and region) were compared across groups using a percentage method to assess chance imbalances following randomisation [51]. Several chance imbalances above 5% points were identified for all five of these variables across the groups and, therefore, gender and age were included as covariates in the models as a sensitivity analysis. It did not make sense to control for SES, region, and education as these were measured with different items across the English and USA samples. Sensitivity analyses were conducted in which covariates were not included into the models, to determine whether the main pattern of results would change (see Appendix A). Model diagnostics (residual plot, Normal P-P plot of residuals) were examined and showed that the regression modelling assumptions were satisfied.

Outliers (±3SDs from the mean) on continuous variables were removed. 47 outliers were removed (2%) from the Encouraging policies variable, and 50 were removed from the BMI variable (2%). There were no other outliers. Sensitivity analyses were conducted in which outliers were not excluded, to determine whether the main pattern of results would change. Data in Figure 1 and Figure 2 were dichotomised (1–4 = 0, 4.01–7 =1) to indicate the proportions of participants that found each policy acceptable (i.e., those rating above the scale midpoint). These dichotomised data are provided to aid interpretation and are not used in any inferential analyses. Cohen’s *d* statistics are covariate-adjusted.

In an exploratory analysis, we used the two one-sided tests (TOST) procedure [56], to evaluate whether our results were equivalent to those reported in the two studies that we were aiming to replicate. Equivalence bounds were set as Δ*_L_* = −0.10 and Δ*_U_* = 0.10 given the size of effects in similar fields. This provided two *p*-values by using t-tests of both below the lower bound (Δ < Δ*_L_*, the lower tail *p*-value) and above the upper bound (Δ*_U_* > Δ, the upper tail *p*-value) with adjusted degrees of freedom using the Sattherwaite method [57]. Equivalence is shown if the largest *p*-value is significant (i.e., data is consistent with being within the two boundaries), and, therefore, only one *p*-value requires reporting. The inference criterion was set at *α* = 0.025 in line with the co-primary analyses.

### 3.2. Results

#### 3.2.1. Policy Support

There were no statistically significant effects of the interventions on the primary outcomes of support for policies. There was no effect of the Group 2 message when compared to the control group on support for encouraging policies, *B* = 0.05, 95% CIs [−0.03, 0.14], *p* = 0.215, *d* = 0.06, or for discouraging policies, *B* = −0.01, 95% CIs [−0.10, 0.08], *p* = 0.823, *d* = −0.01 (see Table 3). The equivalence tests were non-significant for both encouraging policies, *t*(31.22) = 4.35, *p* = 1.00, and discouraging policies, *t*(30.91) = 3.94, *p* = 1.00. This suggests that neither of these two analyses was statistically equivalent to the results of the original study based on equivalence bounds of Δ*_L_* = −0.10 and Δ*_U_* = 0.10 [31] where the effect size was larger *d* = 0.94.

There was also no effect of the Group 3 message when compared to the control group on support for encouraging policies, *B* = 0.06, 95% CIs [−0.03, 0.14], *p* = 0.171, *d* = 0.06, or for discouraging policies, *B* = −0.02, 95% CIs [−0.10, 0.07], *p* = 0.723, *d* = −0.02 (see Table 3). The equivalence tests were non-significant both for encouraging policies, *t*(785.91) = 0.42, *p* = 0.663, and discouraging policies, *t*(737.47) = −0.42, *p* = 0.337. This suggests that neither of these two analyses was equivalent to the results of the original study based on equivalence bounds of Δ*_L_* = −0.10 and Δ*_U_* = 0.10 [32] where the effect size was larger *d* = 0.14.

There were also no statistically significant interaction effects on either of these two policy support outcomes (see Appendix A). This includes two-way and three-way interactions between intervention group, country, and/or BMI.

There was a statistically significant main effect of country on policy support. Participants from the USA reported less support for both sets of obesity prevention policies compared to English participants: Encouraging policies, *B* = −0.15, 05% CIs [−0.22, −0.08], *p* < 0.001, *d* = 0.17; Discouraging policies, *B* = −0.46, 95% CIs [−0.54, −0.39], *p* < 0.001, *d* = 0.48, on the 1–7 rating scale.

#### 3.2.2. Beliefs about the Causes of Obesity (Manipulation Checks)

There was no statistically significant effect of the interventions on the belief that the environment influences obesity, the belief that genetics influence obesity, or the belief that a lack of willpower influences obesity (see Table 4). There were also no statistically significant interaction effects on any of these three causal belief outcomes (see Appendix A). This includes two-way and three-way interactions between intervention group, country, and/or BMI.

There was a statistically significant effect of country on two out of three causal beliefs. American participants were more likely than English participants to believe that genetics influences obesity, *B* = 0.61, 95% CIs [0.51, 0.71], *p* < 0.001, *d* = 0.46, whereas English participants were more likely than USA participants to believe that a lack of willpower influences obesity, *B* = −0.22, 95% CIs [−0.33, −0.11], *p* < 0.001, *d* = 0.15.

### 3.3. Discussion

The aim of Study 2 was to replicate effects of messages previously found in two separate studies to change beliefs and/or attitudes about obesity and obesity policies [31,32]. The results of Study 2 did not replicate these findings. The results of the current study did, however, support the main conclusion reached in Study 1: there is no evidence that communicating information about the environmental causes of obesity changes support for policies to reduce obesity.

We tested two hypotheses to explain why the interventions used in Study 1 did not change support for policies in a manner consistent with previous studies. The first hypothesis was that participants from the USA may be more sensitive to the messages and thus more likely to change their beliefs and attitudes. Study 2 provided no evidence that country of residence (USA vs. England) moderated the effect of messages designed to communicate the impact of environmental causes on obesity on policy support. The second hypothesis, that overweight or obese participants are more likely to change their beliefs and attitudes when presented with information about the environment’s influence on obesity, was also unsupported. There was also no evidence of a three-way interaction effect between BMI, country, and intervention. In summary, there was no evidence that two of the notable differences between Study 1 and previous studies—BMI and country of residence—accounted for the lack of replication. One possible factor explanation for the lack of replication is time; it may be that more people are now aware of the environment’s influence on obesity and therefore ceiling effects are observed.

The results of Study 2 also showed differences in beliefs and attitudes between the USA and English Samples. Participants from the USA, relative to participants from England, were more likely to believe that genetics and less likely to believe that the environment and a lack of willpower caused obesity. Support for policies to reduce obesity was higher in England than in the USA in keeping with existing evidence [52,53].

## 4. General Discussion

The results of two studies suggest that communicating information about the environment’s influence on obesity does not change attitudes towards policies that aim to reduce obesity. This does not replicate earlier studies that reported statistically significant effects using the same messages as those used in the current study [31,32]. Taken together, these two studies suggest that people’s beliefs about the causes of obesity are resistant to change, and in some cases, can even backfire, resulting in people being less likely to believe that the environment influences obesity. Instead, our findings are in line with several studies that have found null effects of similar interventions on policy support [33,34,35]. Targeting people’s beliefs about the causes of obesity is, therefore, unlikely to be a fruitful avenue for increasing support for obesity policies.

There are several factors that could account for the null effect of the messages at changing support for policies. First, the interventions used were ineffective at changing the target beliefs. Although two of the messages changed beliefs in Study 1, one was in the opposite direction to what was hypothesised, and the other was on the non-target belief. The remaining eight of 10 manipulation checks across the two studies failed, suggesting that overall the messages were not sufficiently persuasive or that these beliefs are resistant to change. As similar simple text-based messages comprising evidence or information have been successful at changing other beliefs in related fields (for review see: [44]), it is likely that these beliefs are more resistant to change. A further explanation for the null effects on policy support is that there is a non-causal link between people’s beliefs about the causes of obesity and their support for obesity policies. In the two cases of belief change that were observed in Study 1 there was no subsequent change in support for obesity policies. Although not conclusive, these results are consistent with the explanation that causal beliefs about both general behaviour and obesity-related behaviours do not affect support for obesity policies. Of the previous studies that did change support for obesity polices, only one measured individual obesity causal beliefs, beliefs about the role of affordability, advertising, and the work environment in obesity [32]. The message of interest only changed one of these: it strengthened the belief that food advertising influences obesity. This therefore provides some evidence that changing causal beliefs can increase policy support although the current study did not support these conclusions. 

As the current evidence suggests that beliefs about the causes of obesity are resistant to change, more innovative methods will be needed that can combat the repeated exposure to media, which regularly emphasises that obesity is just a symptom of poor self-control [27,28,29]. Frequently used techniques such as including images, providing evidence, and providing individual narratives have proved ineffectual in the current study and previous research [33,34,35]. One approach that shows promise in other fields is the use of video [58]. Future research could develop video to illustrate the impact of the environment on obesity to determine if this would be successful at changing causal beliefs, and subsequently, if this would lead to greater support for obesity policies.

### Limitations

Study 2 aimed to replicate earlier work by Pearl and Lebowitz [31] and Ortiz, Zimmerman and Adler [32]. However, this study should not be considered a direct replication. We used one intervention from Pearl and Lebowitz [31] and one intervention from Ortiz, Zimmerman and Adler [32], and we used a similar experimental design with a no-message control group. However, there were also several key differences. First, although Ortiz, Zimmerman and Adler [32] used a sample representative of the USA population, as we did in Study 2, Pearl and Lebowitz [31] used a sample solely of people who were overweight or obese. We addressed this by testing whether BMI moderated the effect of intervention, which it did not. Second, our primary and secondary outcomes differed from the outcomes used in Ortiz, Zimmerman and Adler [32] and Pearl and Lebowitz [31]. This was done as several of the policies used in these studies have already been implemented in the UK. The differences in outcome were most notable for Study 1, whereas in Study 2 we used 5/5 of the policies used by [31] and 2/4 of the policies used by [32]. Differences in outcome such as these are common in this literature, and while unlikely to account for the lack of replication, may affect the estimates of effect size. A further limitation is that the images that were added to the messages in Study 1 were not compared against a text & no image group, so it is unclear what effect the addition of these images may have had on the textual messages used in prior studies. A final point is that we used regression on outcomes, which, in some cases, were only on a 1–7 scale, and whilst our model assumes a distribution beyond this range, all modelling diagnostics indicated that this was a reasonable assumption.

## 5. Conclusions

Across two pre-registered studies, comprising samples from the USA and Great Britain, we found no evidence that communicating information on the environment’s influence on obesity-related behaviours increased support for policies designed to reduce obesity. These results do not replicate earlier research that used the same interventions to address this question. Exploratory correlations reported in the Appendix A are consistent with previous studies that identify a relationship between the belief that the environment causes obesity and support for obesity policies. However, the current results suggest that people’s beliefs about the causes of obesity are strongly held and resistant to changing in accordance with evidence. If these beliefs do have a role in support for obesity-related policies, then more persuasive messages would be needed to change beliefs, and, in turn, support for these policies.

## Figures and Tables

**Figure 1 ijerph-17-06539-f001:**
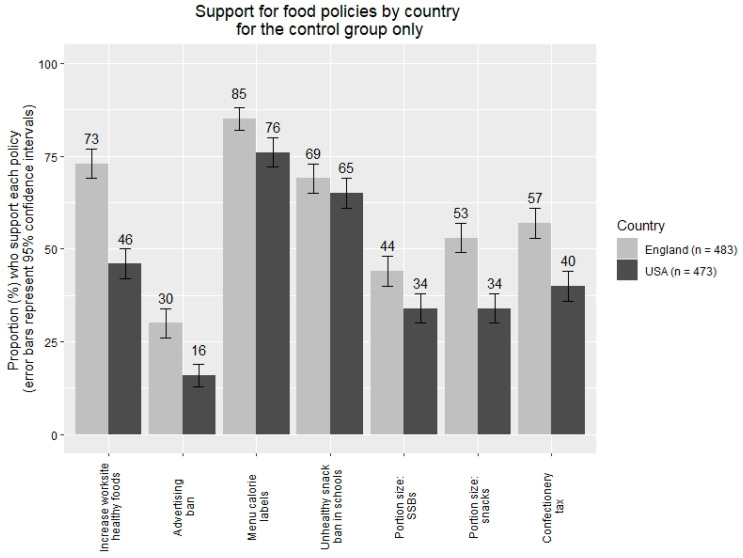
Study 2: Support for obesity policies by country.

**Figure 2 ijerph-17-06539-f002:**
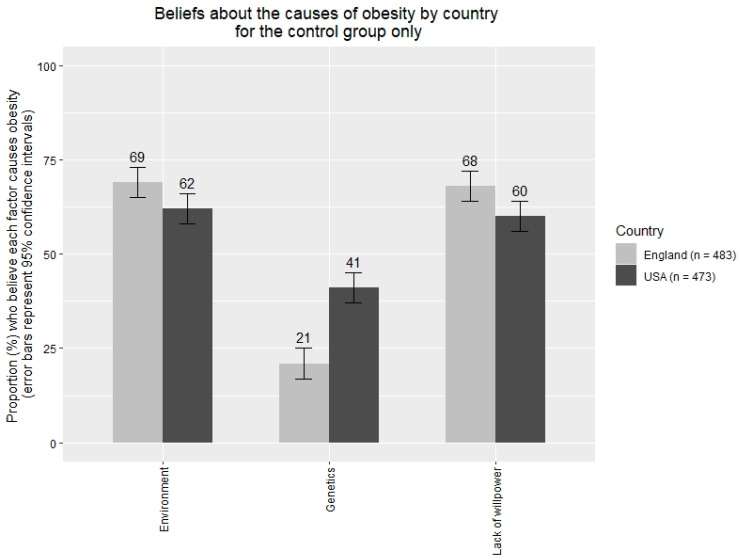
Study 2: Beliefs about the causes of obesity by country.

**Table 1 ijerph-17-06539-t001:** Main effects of intervention messages on support for obesity prevention policies and beliefs about causes of obesity and human behaviour (Study 1).

Variables	Support for Obesity Prevention Policies (*n* = 1680)		Causal Beliefs (Obesity): Environment (*n* = 1680)		Causal Beliefs (Human Behaviour): Environment (*n* = 1680)	
	*B* [95% CIs]	*p*	*B* [95% CIs]	*p*	*B* [95% CIs]	*p*
Age	0.01 [0.00, 0.01]	0.003 *	0.00 [−0.01, 0.00]	0.465	0.00 [−0.01, 0.00]	< 0.001 *
Gender						
Male (ref.)						
Female	0.44 [0.27, 0.61]	< 0.001 *	0.08 [−0.05, 0.20]	0.236	0.10 [0.01, 0.18]	0.023
SES						
DE (ref.)						
C1C2	0.37 [0.15, 0.59]	0.001 *	0.23 [0.07, 0.39]	0.006 *	0.20 [0.09, 0.31]	< 0.001 *
AB	0.51 [0.26, 0.76]	< 0.001 *	0.26 [0.07, 0.44]	0.007 *	0.27 [0.15, 0.39]	< 0.001 *
Education						
Low (ref)						
Medium	0.06 [−0.15, 0.27]	0.571	−0.03 [−0.19, 0.13]	0.704	0.21 [0.10, 0.31]	< 0.001 *
High	0.17 [−0.07, 0.41]	0.161	0.07 [−0.11, 0.25]	0.432	0.35 [0.23, 0.46]	< 0.001 *
Region						
London (ref)						
North East	0.04 [−0.44, 0.51]	0.883	−0.08 [−0.43, 0.27]	0.667	0.11 [−0.12, 0.35]	0.346
North West	−0.14 [−0.49, 0.22]	0.451	−0.14 [−0.41, 0.12]	0.282	−0.08 [−0.25, 0.09]	0.376
Yorkshire & Humb.	0.16 [−0.20, 0.52]	0.376	−0.23 [−0.49, 0.04]	0.094	−0.01 [−0.19, 0.16]	0.874
East Midlands	−0.04 [−0.43, 0.36]	0.852	−0.01 [−0.30, 0.28]	0.938	0.05 [−0.14, 0.24]	0.604
West Midlands	0.01 [−0.35, 0.37]	0.964	−0.18 [−0.45, 0.09]	0.183	−0.13 [−0.30, 0.05]	0.160
East of England	0.13 [−0.25, 0.50]	0.510	−0.15 [−0.42, 0.13]	0.294	−0.10 [−0.29, 0.08]	0.261
South East	−0.10 [−0.43, 0.24]	0.564	−0.15 [−0.39, 0.10]	0.248	−0.12 [−0.28, 0.05]	0.165
South West	−0.03 [−0.38, 0.32]	0.856	−0.18 [−0.44, 0.08]	0.167	0.13 [−0.04, 0.31]	0.121
Wales	−0.06 [−0.51, 0.39]	0.785	−0.18 [−0.51, 0.15]	0.288	−0.12 [−0.34, 0.10]	0.294
Scotland	−0.33 [−0.70, 0.04]	0.079	−0.47 [−0.75, −0.20]	0.001 *	−0.21 [−0.39, −0.02]	0.028
Group						
Control (ref)						
Obesity message (a)	0.08 [−0.19, 0.35]	0.559	−0.01 [−0.21, 0.19]	0.922	0.18 [0.04, 0.31]	0.009 *
Obesity message (b)	−0.14 [−0.41, 0.13]	0.314	−0.27 [−0.47, −0.07]	0.007 *	−0.03 [−0.16, 0.10]	0.669
Behavioural message (a)	−0.04 [−0.31, 0.22]	0.756	−0.09 [−0.29, 0.10]	0.345	0.05 [−0.08, 0.18]	0.482
Behavioural message (b)	0.04 [−0.23, 0.30]	0.783	−0.02 [−0.21, 0.17]	0.829	0.10 [−0.03, 0.22]	0.138

* *p* < 0.0125.

**Table 2 ijerph-17-06539-t002:** Study designs.

Group	Study 1	Study 2
Group 1	Control (no message)	Control (no message)
Group 2	Obesity message (a) + images	Obesity message (a) (no image)
Group 3	Obesity message (b) + images	Obesity message (c) (no image)
Group 4	Behaviour message (a) + images	
Group 5	Behaviour message (b) + images	

**Table 3 ijerph-17-06539-t003:** Main effects of intervention messages on support for obesity prevention policies (Study 2).

Variables	Support for Encouraging Policies (*n* = 2544)		Support for Discouraging Policies (*n* = 2586)	
	*B* [95% CIs]	*p*	*B* [95% CIs]	*p*
Age	0.00 [0.00, 0.00]	0.728	0.00 [0.00, 0.00]	0.590
Gender				
Male (ref)				
Female	0.23 [0.16, 0.30]	< 0.001 *	0.26 [0.19, 0.34]	< 0.001 *
Group				
Control (ref)				
Obesity (a)	0.05 [−0.03, 0.14]	0.215	−0.01 [−0.10, 0.08]	0.823
Obesity (c)	0.06 [−0.03, 0.14]	0.171	−0.02 [−0.10, 0.07]	0.723
Country				
England (ref)				
USA	−0.15 [−0.22, −0.08]	< 0.001 *	−0.46 [−0.54, −0.39]	< 0.001 *

* *p* < 0.025.

**Table 4 ijerph-17-06539-t004:** Main effects of intervention messages on beliefs about the causes of obesity (Study 2).

Variables	Causal Beliefs: Environment (*n* = 2711)		Causal Beliefs: Genetics (*n* = 2711)		Causal Beliefs: Willpower (*n* = 2711)	
	*B* [95% CI]	*p*	*B* [95% CI]	*p*	*B* [95% CI]	*p*
Age	−0.01 [−0.01, −0.01]	< 0.001 *	0.00 [0.00, 0.00]	0.317	0.00 [0.00, 0.01]	0.143
Gender						
Male (ref)						
Female	0.27 [0.15, 0.38]	< 0.001 *	0.11 [0.01, 0.21]	0.034	−0.33 [−0.45, −0.22]	< 0.001 *
Group						
Control (ref)						
Obesity (a)	−0.12 [−0.26, 0.02]	0.087	−0.04 [−0.16, 0.08]	0.519	0.11 [−0.03, 0.24]	0.120
Obesity (c)	−0.14 [−0.28, 0.00]	0.053	0.00 [−0.12, 0.13]	0.956	0.07 [−0.07, 0.20]	0.331
Country						
England (ref)						
USA	−0.15 [−0.26, −0.03]	0.011	0.61 [0.51, 0.71]	< 0.001 *	−0.22 [−0.33, −0.11]	< 0.001 *

* *p* < 0.01.

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
