# Peer review of "Communicating Evidence about the Causes of Obesity and Support for Obesity Policies: Two Population-Based Survey Experiments"

_ijerph, 2020, doi:10.3390/ijerph17186539_

Round 1

Reviewer 1 Report

This paper presents the results of two separate studies exploring the effect of messaging on support for obesity-related policies. The sample sizes of both studies were commendable and the authors followed appropriate protocols for registering the studies in advance.

In general, I have trouble with the presentation of the two studies in one paper. The original idea to use Study 2 to further test the results of Study 1 didn't work out due to the results of Study 1, so the two are distinct. There are inconsistencies with the way the methods are presented across the two studies within the paper (e.g. Box 1; underlining in lines 288-290 with more details about messaging). There is limited justification provided for the comparisons between England and the US. The authors may want to consider splitting the two studies into two papers, or at a minimum presenting the methods of the two studies in the same way.

My other concern focuses on how the work is situated within the previous evidence. While I appreciate the contrast to the literature (e.g. discussion for study 1), in the background section you presented the evidence as mixed (e.g. line83 "Given the mixed results..."). This was the major rationale for the study. Later you seem to be presenting the previous evidence as different from your results (e.g. line227 "...was unexpected.."). These two interpretations seem at odds. The same issue repeats in the discussion and limitation sections of study 2. 

Some other smaller comments:

Background:
-first parag: clarify whether you are limiting to policy strategies. The second sentence mentions policy, but other language is used that may include non-policy strategies
-line42-43 missing end of parentheses

-in general, the background on attribution theory is difficult to follow because so many different terms are used interchangeably. It might improve clarity to define AT and then provide examples, and to use the same language to describe the theory you are using (e.g. attribution theory, fundamental attribution error, correspondence bias, causal attributions, causal beliefs)

Table 1: Unclear why so many different regions were included (this was not described in methods); footnote should read p<0.0125

Study 2 method: 

-why were different inclusion criteria used (quotas) for the two countries?

-Provide more details about how the seven policy questions were combined into the two measures. This is important to understanding the discussion on outliers.

Reviewer 2 Report

In general, the study deals with an interesting topic. The interaction is well-written and the sample size is wide enough to ensure strong results. However, the applied methodology is not adequate to the type of data and this prevents the observation of any result.

Revision should be made particularly on the statistics part (managing of data and analyses) to provide detailed information. It is not sufficient to drop some quick sentences and refer to previous literature, the readers should be able to understand what the author did without going to read previous papers. Most importantly, the methodologies should be changed according to the type of data (e.g. OLS regression cannot be applied on non-continuous data). More comments are provided below.

Background

  • I am not familiar with the literature on support for obesity policies, but the introduction seems well written to me. It is clear and enjoyable to read. The only suggestion is to include some information about the distinction between internal and external causes for attribution, as explained in Kelley’s paper. I believe the authors refer to internal attribution, but it is not specified in the text.

Study 1

  • The exact effect size from Ortiz, Zimmerman and Adler [32] could not be calculated; however, a power calculation suggested that this sample size would provide 80% power to detect small effects between two groups (d = .26) after a Bonferroni adjustment (α = .0125): what is the “exact effect size”? I found no reference to it in the mentioned paper. Why could not be calculated? What is specifically “a power calculation?” What is d in parentheses? The whole paragraph is unclear. The author should specify in detail how they calculated the effects.
  • “After applying weighting, mean age = 48.33 (SD = 16.87) and 51.6% were female”: which weighting? There is no reference to weighting before. Please specify which kind of weighting has been applied.
  • Groups in the online OSF have different numbers with respect to supplementary material and the paper, this creates confusion.
  • It is no clear what “marketing” means and adds to the messages, how do messages with and without marketing differs? This should be explained in the text.
  • How are the policy presented? Include the questionnaire text in supplementary material
  • Lines 172-176 are about participants and not on analyses, please revise the structure to ensure a more logical development (here the authors refer to weights, which were previously mentioned above without explanation)
  • Provide more information on “a percentage method to assess chance imbalances”, what is this?
  • Provide results of all Sensitivity analyses, also with included outliers. What the authors call outliers are likely to be real extreme values, as they are likert-type response provided by respondents. I am not sure these should be excluded.
  • Note in table 1 is .0125
  • If I am correct, the authors use OLS regression with dependent variable policy support on a likert scale 1-7. This is not a continuous variable, and OLS regression is formally wrong. Ordered logit model can be used instead, or logit model after merging the likert scale in two categories. Moreover, difference among treatments cannot be found if all the treatments are taken together. The author could use Kruskal-Wallis tests to analyze differences between treatments on a likert-type response (see Biondi and Camanzi, 2020)

Study 2

  • Some of the comments above are also applied to study 2. I suggest the authors perform a deep revision of the statistical methodologies applied
  • I suggest harmonization in the presentation of studies between study 1 and 2. For instance, removing the box in Study 2 and presenting messages in the supplementary material.
  • Looking at the factor loadings of the PCA, it is not so clear the difference between encouraging and discouraging policies, since there is a ban in the first category. I suggest to use a categorical-pca, which deals with categorical data like likert type responses (see Linting et al 2007)
  • Here, OLS can be used since the dependent variable is not the categorical likert type answer, but the PCA result of items. Still, I think that Kruskal-Wallis tests can provide better results on treatment effect alone, before analyzing several effects together (e.g. demographics also).
  • I am still not convinced that outliers should be removed. The results of no effect could also depend on the removal of extreme data points.

References

Biondi, B., & Camanzi, L. (2020). Nutrition, hedonic or environmental? The effect of front-of-pack messages on consumers’ perception and purchase intention of a novel food product with multiple attributes. Food Research International, 130, 108962.

Linting, M., Meulman, J. J., Groenen, P. J., & van der Koojj, A. J. (2007). Nonlinear principal components analysis: introduction and application. Psychological methods, 12(3), 336.

Round 2

Reviewer 1 Report

I still do not feel that combining the two studies in one paper makes sense. While I appreciate the additions to the paper, I also have remaining questions (indicated in my initial review) about the methods and statistics that make the results difficult to properly interpret. Much of the discussion provided in the authors' response to previous comments should be integrated into the paper itself, rather than explained to the reviewers. 

Author Response

#Reviewer 1.

I still do not feel that combining the two studies in one paper makes sense.

Response: Without further explanation from the reviewer as to why this does not make sense, we cannot directly address their concerns. In our last resubmission we responded to their initial concerns by ensuring that the two studies were reported in a similar way. While reporting two studies within one paper may not be familiar to the reviewer, this is quite common practice and has numerous advantages. For example, results of individual studies sometimes do not replicate, and are cited heavily until a later replication attempt is done which contradicts the original study. Conducting and reporting two studies together as we have done provides significantly stronger evidence than a single study.

Although the two studies in our paper are not identical, they do overlap in the most important way: they both test our main research question. Namely, does communicating evidence of the environment’s influence on obesity increase support for policies that aim to tackle obesity? These two studies converge on the conclusion that this evidence may not be sufficient to change public support.

While I appreciate the additions to the paper, I also have remaining questions (indicated in my initial review) about the methods and statistics that make the results difficult to properly interpret. Much of the discussion provided in the authors' response to previous comments should be integrated into the paper itself, rather than explained to the reviewers. 

Response: It is not clear which aspects of our previous response the reviewer wants us to add into the paper. Looking through our previous responses there was only one response we made to the reviewer that did not also come with a change to the manuscript. This was the following:

 “-why were different inclusion criteria used (quotas) for the two countries?”

We therefore have added our explanation into the paper. p9, line 290-291.

            “These were different for the two countries as the research agency had different methods for ensuring a representative sample for each country.”

We also reviewed our other responses to see if more details in the methods and statistics should have been added. The only comment we believe the reviewer could be referring to is our response to how we combined the 7 items into the primary outcome for study 2, as this was the reviewer’s only comment that addressed statistics:

-Provide more details about how the seven policy questions were combined into the two measures. This is important to understanding the discussion on outliers.

Our changes to the manuscript following this comment was to review reporting guidelines for PCA and to add in this information. Reviewing our response to this question, we believe the only improvement that could be made is that we did not specify the minimum KMO (Kaiser-Meyer-Olkin) value that is needed to ensure the PCA was valid, i.e. .50. We have therefore added this in.

“The Kaiser-Meyer-Olkin measure verified the sampling adequacy (KMO = .85; Bartlett’s test < 0.01), well above the minimum .50 that is needed [53].”

Reviewer 2 Report

I am satisfied with the authors replies to my earlier comments.

Author Response

Thank you for your positive comments